# Identification of Urban Functional Areas and Urban Spatial Structure Analysis by Fusing Multi-Source Data Features: A Case Study of Zhengzhou, China

**Jinxin Wang, Chaoran Gao \*, Manman Wang and Yan Zhang**

School of Geoscience & Technology, Zhengzhou University, Zhengzhou 450001, China
\* Correspondence: gcr1277qcym@163.com

**Abstract:** The identification and delineation of urban functional zones (UFZs), which are the basic units of urban organisms, are crucial for understanding complex urban systems and the rational allocation and management of resources. Points of interest (POI) data are weak in identifying UFZs in areas with low building density and sparse data, whereas remote sensing data lack the necessary semantic information for functional zoning, and single-source data cannot perform a highly comprehensive characterization of complex UFZs. To address these issues, this study proposes a method for identifying UFZs by fusing multi-attribute features from multi-source data and introduces nighttime light and land surface temperature (LST) indicators as functional zoning references, taking the main urban area of Zhengzhou as an example. The experimental results show that the POI data with integrated three-level semantic information can characterize the semantic information of functional areas well, and the incorporation of multi-spectral, nighttime light, and LST data can further improve the recognition accuracy by approximately 10.1% compared with the POI single-source data. The final recognition accuracy and kappa coefficient reached 84.00% and 0.8162, respectively, indicating that the method is largely consistent with the actual situation and is feasible. The analysis showed that the main urban area of Zhengzhou as a whole is characterized by the coordinated development of single and mixed functional areas, in which a distinct residential-commercial-public complex is formed, and the urban functional areas on the block scale have diverse attributes. This study can provide a decision-making reference for the future development planning and management of Zhengzhou, China.

**Keywords:** functional zoning; multi-source data; POI; focal loss; LightGBM; urban spatial structure



## 1. Introduction

With rapid progress of the information age, sensing technologies and computing environments have undergone profound advances and improvements. Consequently, means of obtaining data, such as urban sensing, are becoming increasingly abundant. As geographical products, cities form the core of economic and social development. Urban functional zones (UFZs) are important geospatial attributes of urban land, a geographical space where social resources are gathered and specific urban functions are effectively performed [1]. UFZs are both relatively independent and interconnected, forming an organic urban whole [2,3]. The accurate identification of UFZs and the elucidation of their distribution patterns are essential for promoting modern urban construction management and for optimizing urban industrial structures; however, the accurate identification of UFZs is challenging owing to the complexity and comprehensiveness of urban functions [4].

Most traditional studies rely on existing land use information, field surveys, expert evaluations, or thematic data for functional zoning, which are subjective, lack objective tests, and are time-consuming and labor-intensive [5–7]. In recent years, high-quality very-high-resolution remote sensing images, with their large coverage, rich image information, and wide availability, have demonstrated certain advantages in representing UFZs [8].

However, owing to the lack of semantic information on UFZs, their identification is limited by certain factors [9]. For example, remotely sensed images comprise visual features of functional partitions but cannot provide object information. Missing object information in turn leads to both the object categories and spatial object patterns being ignored, which may lead to inaccurate classification results. Data types have gradually diversified with the development of information and perception technologies. The use of points of interest (POI) [10], social media check-ins [11], floating vehicle ODs [12], and mobile phone signaling data [13,14], which can characterize geographic location information and semantic information related to human activities, yields better results for urban functional zoning [15] or dynamic description [16] and can fill the gaps in missing semantic information regarding the functional space of remote sensing images.

The combination of high-resolution remote sensing images and social perception data is an effective way to identify urban functional areas quickly. However, only one type of attribute can describe the characteristics of urban functional areas at a certain level. To entirely reflect the characteristics of urban functional areas, multidimensional characteristic information must be obtained from various sources. In contrast to daytime remote sensing, nighttime remote sensing images record information on artificial lights, which can reflect the distribution of human settlements and the intensity of socioeconomic activities [17], which are strongly related to the spatial structure of cities [18]. Huang et al. [19] proved that light information related to human activities captured by nighttime light remote sensing data can provide additional useful information for urban functional zones interpretation. However, the heterogeneity of urban areas is not only reflected in the heterogeneity of spatial landscapes and the differentiation of socio-economic functions but is also related to the imbalance of anthropogenic heat emissions in urban areas. As land surface temperature (LST) is highly influenced by human social activities [20], studies have been conducted to analyze the relationship between LST changes and the structural characteristics of different functional areas [21]. Feng et al. [22] demonstrated that different urban functional area types have different LST effects. Therefore, nighttime light and LST data provide a unique perspective for identifying UFZs.

A city's spatial structure and layout are concrete representations of its functional area. The study of spatial structure can provide a scientific basis for the rational deployment of resources in functional areas, urban spatial planning, and the promotion of urban development [23]. The fusion of multi-source data is currently an effective method to identify UFZs rapidly and accurately, providing an understanding of the spatial layouts of cities. However, most studies use single-source POI data [24,25] for the identification and analysis of UFZs, which often only express the socio-semantic characteristics of UFZs at a single level and cannot comprehensively reflect their geovisual and socioeconomic characteristics. To provide a comprehensive characterization of complex UFZs, Zhang [26] and Chang [27] used random forest (RF) to fuse multi-spectral and social perception data for functional zoning. However, the issue of spatial sparsity of social perception data requires further exploration. Chen [28] used a light gradient boosting machine (LightGBM) to fuse remote sensing images and user behavior data to identify UFZs; however, because of the use of fragmented remote sensing images within the study, the results did not directly reflect the distribution of different UFZ types in the city and their interrelationships. Additionally, Huang [19] fused daytime and nighttime remote sensing data for functional zoning, demonstrating that nighttime light features can effectively complement daytime remote sensing data features. Li [29] used RF to fuse LST and urban morphological features to classify UFZs, thereby demonstrating that LST can also be used to characterize different UFZs. High-resolution remote sensing imagery can provide high-resolution and large-scale descriptions of UFZs that can provide visual features, such as spectra, textures, and geometries, which can aid in their identification. POI data and nighttime light imagery can provide socio-economic features for the identification of urban functional areas, and LST can be used to describe different UFZs. However, fully combining these data characteristics and information to identify urban functional areas is challenging.

The recent and rapid increase in the use of machine learning (ML) techniques in digital image processing applications has led to renewed interest in satellite image classification [30]. Support vector machines (SVM) and ensemble classifiers, such as RF, have been widely utilized for land use and land cover (LULC) classification owing to their ability to handle high-dimensional data and to perform well with limited or unbalanced training samples [31]. Furthermore, these previous studies have demonstrated that the utilization of aggregated classifiers generally provides an improved classification performance compared to individual weak classifiers [32]. Examples include canonical correlation forests (CCF) [33], extreme gradient boosting (XGBoost) [34], LightGBM [28], and categorical boosting (CatBoost) [35]. McCarty et al. [36] demonstrated the superiority of LightGBM over popular ML algorithms such as RF and SVM in large-scale LULC classification studies. In an evaluation study of crop classification, Ustuner et al. [37] reported that LightGBM showed superior classification performance, and the required processing time was very short compared to that of XGBoost. Colkesen [30] demonstrated that the LightGBM algorithm had the highest classification performance for medium spatial resolution Sentinel-2 images. In light of the latter, the present study uses LightGBM and introduces the Focal Loss (FL) function to construct the FL-LightGBM algorithm which fuses multimodal data for UFZ classification. Additionally, the FL function is introduced to address the imbalance problem that exists in functional area type samples.

Based on the above foundation and problems highlighted, this study fully combines spatial visual, socioeconomic, and LST feature information, highlights the rich semantic information contained in POI data, constructs three-level semantic indicators to express the differences between different types of functional areas, and provides a comprehensive characterization of complex UFZs. The FL-LightGBM algorithm was used to fuse multimodal data for the UFZ classification. Subsequently, the classification results were reconstructed at the block scale of the OpenStreetMap (OSM) road network segmentation to obtain the distribution of functional areas in the block units, with a detailed delineation of single and mixed functional areas. Finally, a spatial structure analysis method was used to further analyze the distribution pattern characteristics of the UFZs. This study can be used as a reference for identifying urban functional areas and for planning the functional layout structures of the cities.

## 2. Study Area and Data Sources

### 2.1. Study Area

Zhengzhou, the capital city of Henan Province, is situated north-centrally at the boundary between the middle and lower reaches of the Yellow River (112°42′ E–114°14′ E, 34°16′ N–34°58′ N). As a national central city in China, Zhengzhou has distinct location advantages and is one of the key cities in "the Belt and Road" region; it is also an important hub for national transportation, logistics, communication, and commercial trade, with an important role in gathering and radiation. According to data published by the Statistics Bureau of Zhengzhou (http://tjj.zhengzhou.gov.cn/ (accessed on 21 August 2022)), the population of Zhengzhou will reach 12,742,000 by the end of 2021, and its gross domestic product will reach USD 184,102 million. As this study focused on urban functional zones, the main urban area within the Fourth Ring Road of Zhengzhou was selected as the study area (Figure 1). The Fourth Ring Road of Zhengzhou is 93.3 km long, and its surrounding area contains five administrative districts: Jinshui, Zhongyuan, Guancheng, Huiji, and Erqi.

### 2.2. Data Source and Processing

#### 2.2.1. Sentinel-2 Data

The Sentinel-2 high-resolution multi-spectral imagery data covered 13 spectral bands with a spatial resolution of up to 10 m. Level 1C products were downloaded from the official website of the United States Geological Survey (USGS) (https://earthexplorer.usgs.gov/ (accessed on 14 July 2022)) for 7 April 2022. Images from early spring have low vegetation shading but possess a certain degree of recognizability. These data were atmospherically

corrected using the Sen2cor tool, and the band with a spatial resolution of 20 m was re-sampled to 10 m using the SNAP software. Images were then subjected to band fusion and multi-scale segmentation using the eCognition software. The purpose of multi-scale segmentation is to obtain patches of target features, obtain information on the various data characteristics of the patches as a unit for the classification of urban functional areas, and finally to reconstruct them at the block scale. This experiment primarily involved parameters such as the segmentation scale, shape and compactness. After several experiments, we found that the best results were obtained by setting the weight of each band to 1, the scale to 60, the shape to 0.5, and the compactness to 0.4.

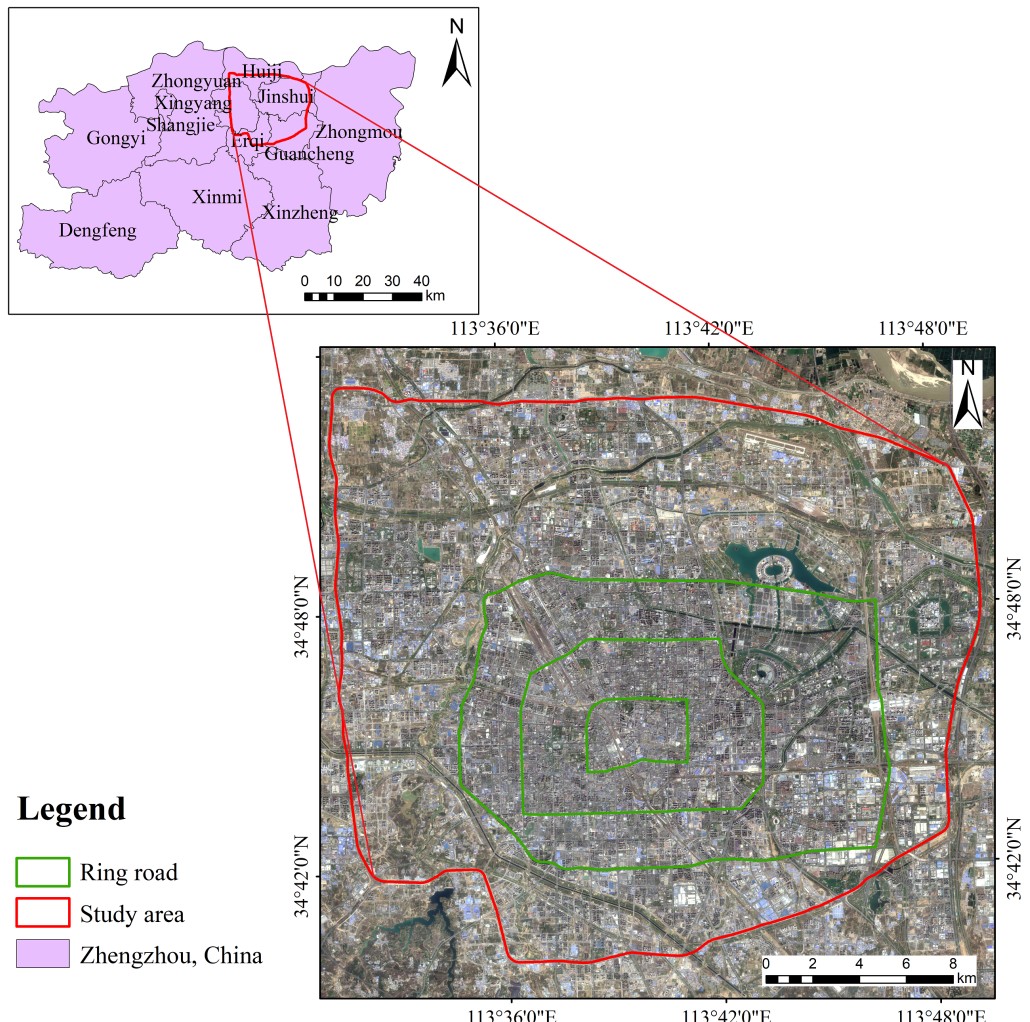

**Figure 1.** Extent of the study area. The data source of the remote sensing image base map is Sentinel-2.

### 2.2.2. NPP-VIIRS Nighttime Light Data

NPP-VIIRS nighttime light data were obtained from the National Oceanic and Atmospheric Administration (NOAA), and the acquisition product was a monthly product for April 2022 with a spatial resolution of 500 m. Information on the characteristics of these data was obtained by counting the brightness values in the patch range (please refer to Section 3.1 for specific characteristic information).

### 2.2.3. POI Data

The POI data for Zhengzhou were obtained from the AutoNavi Maps Open Platform (https://lbs.amap.com/ (accessed on 14 May 2022)) and Baidu Maps Open Platform (https://lbsyun.baidu.com/ (accessed on 14 May 2022)), with 222,801 items after cleaning and deduplication. Each POI data record includes attribute information, such as a name,

address, category, latitude, and longitude. POI data have a wide range of classification categories and their classification standards are inconsistent with those for urban land use. Therefore, data with a low degree of public awareness were deleted and POI data were reclassified according to the Code for Classification of Urban Land Use and Planning Standards of Development Land (GB50137-2011) and the Baidu Maps/AutoNavi Maps POI classification standard. The types of UFZs and reclassification of POIs are listed in Table 1.

**Table 1.** Types of urban functional zones (UFZs) and point of interest (POI) reclassification table.

| Code | Primary Classification | Secondary Classification | Tertiary Classification |
|---|---|---|---|
| R | Residential land | Residential district | Villas, communities, neighborhoods |
| B | Commercial land | Catering services<br>Shopping services<br>Accommodation services<br>Financial and insurance services<br>Commercial land<br>Life services<br>Other facility services | Restaurants, dessert shops, beverage shops<br>Shopping malls, convenience stores, shopping streets<br>Hotels, guesthouses, accommodation services<br>Banks, ATMs, insurance companies, other financial and insurance service providers<br>Companies, business offices<br>Information and consultation center, ticket offices, post offices, and telecom business office<br>Motorcycle service, automobile services, filling station, gas station, charging station |
| M | Industrial land | Industrial land | Factory, farming, forestry, animal husbandry, fishery base, industrial park |
| A | Administration and public services land | Science/culture and education services<br>Sports and recreation services<br>Medical services<br>Governmental organizations and social group<br>Public facilities | School, media organization, research institution, training institution, museum, library, archives<br>Sports stadium, recreation center, recreation space<br>Hospital, clinic and medical and health care services<br>Governmental organization, social group, public security organization and industrial and commercial taxation institution<br>Newsstand, emergency shelter |
| S | Transport land | Transportation services | Airport-related, railway station, coach station, toll gate, service area, parking lot, subway station |
| G | Scenic areas and squares land | Park squares | Park, square, city plaza |
| E | Other land | Cultivated land, forest, grassland, non-construction land | |

### 2.2.4. Landsat 8-9 OLI/TIRS Data

Landsat 8-9 data were sourced from the USGS website (https://earthexplorer.usgs.gov/ (accessed on 16 August 2022)). The acquired products comprised Landsat 8 and Landsat 9 Collection 2 Level-2 surface temperature products with less than 10% cloud cover from 1 April to 1 May 2022, with a total of three images. The temperature products were converted to Celsius using the conversion formula, after which they were averaged to obtain monthly data.

### 2.2.5. OSM Road Network Data

The road network vector data used in this study were derived from OSM (https://www.openstreetmap.org/ (accessed on 12 April 2022)), including motorway, primary, secondary, tertiary, and unclassified (in mainland China, this classification may correspond to a neighborhood road or village road in urban planning), and residential roads at multiple levels, for a total of 11,169 roads.

### 2.2.6. Administrative Division Data

Administrative division data were downloaded from the National Geomatics Center of China (NGCC) (http://www.ngcc.cn/ngcc/ (accessed on 14 May 2022)).

The year of the data representations was 2022. For research and computational purposes, the raster data used in this study were uniformly re-projected and resampled.

## 3. Methodology

The workflow of the method proposed in this study comprises four processes (Figure 2): (1) multi-scale feature segmentation using Sentinel-2 data to obtain feature patches of different scale sizes; (2) feature extraction of Sentinel-2, POI, nighttime light, and Landsat 8-9 data at the feature patch scale; (3) using FL-LightGBM to fuse multiple source data features for model training and learning to complete recognition and classification; and (4) reconstruction of the identification results at the block scale, detailed division of the functional urban land use, and the overall layout and analysis.

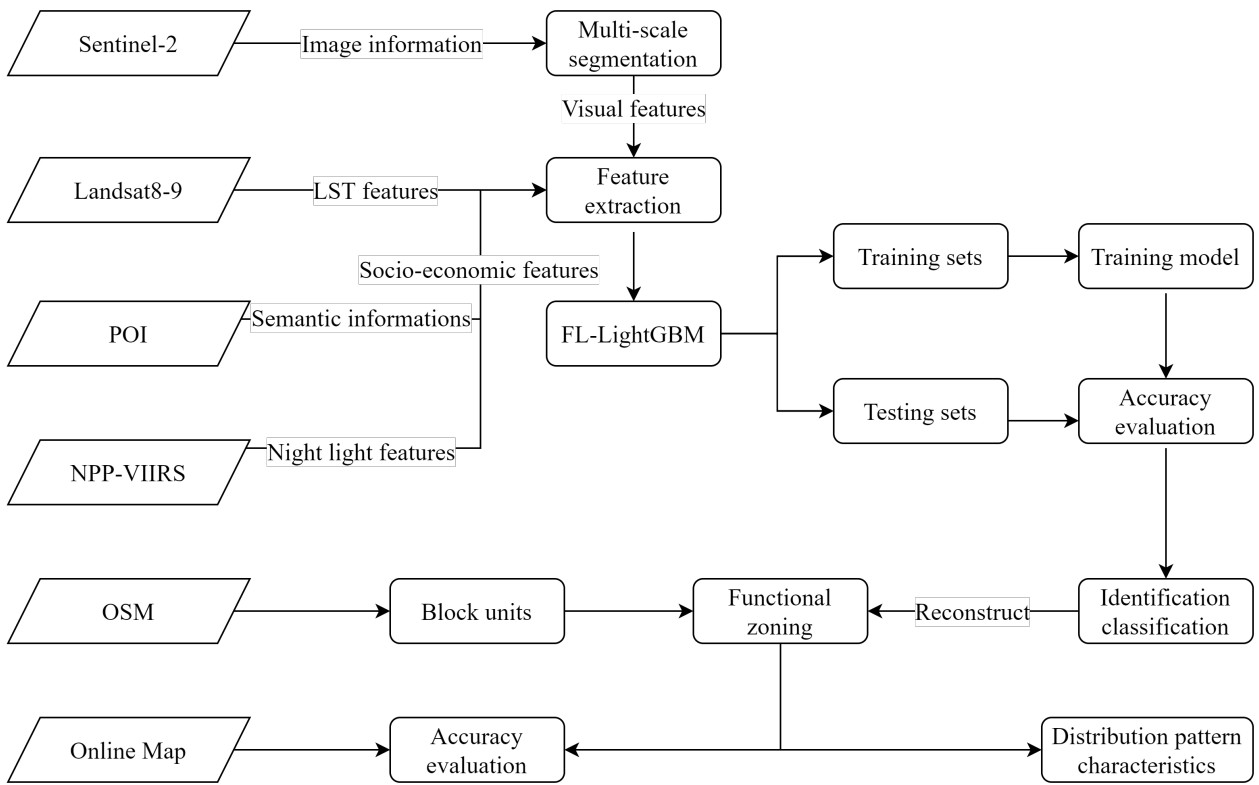

**Figure 2.** Flowchart for the study of the proposed method.

### 3.1. Multi-Feature Extraction for Multi-Source Data

3.1.1. Sentinel-2 Remote Sensing Image Feature Extraction

Image feature extraction includes the feature extraction of multi-scale segmented objects, including spectral, textural, and shape feature information. The normalized difference built-up index (NDBI), normalized difference water index (NDWI), and normalized difference vegetation index (NDVI) were obtained using band operations (Equations as (1)–(3)). Other spectral, texture, and shape features were calculated using the eCognition software, and the average value of each feature was obtained as the feature indicator.

$$NDBI = \frac{B11 - B8}{B11 + B8} \tag{1}$$

$$NDWI = \frac{B3 - B8}{B3 + B8} \tag{2}$$

$$NDVI = \frac{B8 - B4}{B8 + B4} \tag{3}$$

### 3.1.2. POI Data Feature Extraction

As the number of POI data points varies considerably, and the semantic information contained in each type of POI has a different degree of influence on the functional definition of urban land, the weights were determined for each type of POI data according to the following reclassified three-level categories to accurately characterize the functional semantic information represented by each POI.

Considering that the number of each POI type had a different degree of impact on the functional zoning of the city, the primary weight values ($W_1$) were determined using the paired factor comparison method (Equation (4)).

$$W_i = \frac{\sum_{j=1}^{N} W_{ij}}{N} \tag{4}$$

where $N$ is the number of primary categories; $W_{ij}$ is the ratio of category $i$ after two comparisons between categories $i$ and $j$; and $W_i$ is the primary weight of category $i$.

The indicator of the public cognition degree [38] was cited as another influence factor based on the significant cognition of various types of POIs, and secondary weight values ($W_2$) were determined with reference to the literature [38,39].

The POI as point data does not contain the area information of geographic entities, and the areas of different geographic entities vary considerably. Considering the land-use scale represented by each type of POI, the average area of the tertiary classification POI was used to determine the tertiary weight values ($W_3$) according to the national business classification standard with reference to the relevant literature [40,41].

The three levels of category weights were combined to reflect the impact of the category on the definition of the functional area clearly and to determine the final weights: $W_i = W_1 * W_2 * W_3$. Finally, the data were standardized to eliminate cases in which the total **score** for a particular type of POI was minimal and to reduce the impact of distributional differences on the model.

The average weighted kernel density values of the various types of POI and the total POI data were calculated as the POI semantic features.

### 3.1.3. Nighttime Light Data Feature Extraction

The maximum value, standard deviation, average value, and difference between the maximum and minimum intensity values of nighttime light were used as nighttime light data feature indicators.

### 3.1.4. LST Feature Extraction

Processed Landsat 8-9 OLI/TIRS data were used to calculate the maximum value, standard deviation, mean value, and difference between the maximum and minimum values of LST as LST feature indicators.

A total of 43 features were extracted based on Sentinel-2, POI, nighttime light, and Landsat 8-9 OLI/TIRS data, which are listed in Table 2.

The calculated individual data features are shown in Figure 3, which displays the six statistical values for each of the 43 data features, namely the upper quartile, median, mean, lower quartile, minimum, and maximum, after removing outliers, and showing the variation and diversity in the visual, socioeconomic, and LST features of the urban space. Most data features exhibit a relatively symmetrical median, indicating fluctuations within a relatively stable range, and the presence of outliers supports the regional classification. Certain data characteristics exhibited a trend with a minimum value of zero, resulting in high outliers and indicating a large variation in such data characteristics between regions. Therefore, these described data features can effectively characterize the visual, socio-economic, and LST features of a city, and they can be used to classify different urban functions.

**Table 2.** Description of each type of extracted feature. NDWI, normalized difference water index; NDVI, normalized difference vegetation index; NDBI, normalized difference built-up index; POI, point of interest; LST, land surface temperature.

| Data Sources | Features | Description of the Features | Name of the Features |
|---|---|---|---|
| Sentinel-2 | Spectral features | NDWI, NDVI, NDBI | NDWI, NDVI, NDBI |
| | | Mean Blue2, Mean Green3, Mean Red4, Mean NIR8, Mean SWIR11 | Mean_B2, Mean_B3, Mean_B4, Mean_B8, Mean_B11 |
| | | Standard deviation Blue2, Standard deviation Green3, Standard deviation Red4, Standard deviation NIR8, Standard deviation SWIR11 | St_B2, St_B3, St_B4, St_B8, St_B11 |
| | | Skewness Blue2, Skewness Green3, Skewness Red4, skewness NIR8, skewness SWIR11 | Sk_B2, Sk_B3, Sk_B4, Sk_B8, Sk_B11 |
| | Texture features | Gray-level co-occurrence matrix (GLCM) mean, GLCM entropy, GLCM contrast, GLCM correlation, GLCM dissimilarity | GLCM_Mean, GLCM_Entropy, GLCM_Contrast, GLCM_Correlation, GLCM_Dissimilarity |
| | Shape features | Length/width, density, mean shape index, length and area of the shape | Length/Width, Density, Shape_index, Shape_Length, Shape_Area |
| POI | POI features | Average weighted kernel density values of the various types of POI and the total POI | R_Kernel, B_Kernel, M_Kernel, A_Kernel, S_Kernel, G_Kernel, ALL_Kernel |
| NPP-VIIRS nighttime light | Nighttime light features | Maximum value, standard deviation, average value and the difference between the maximum and minimum values of the intensity values of nighttime light | DN_Max, DN_Std, DN_Avg, DN_Range |
| Landsat 8-9 OLI/TIRS | LST features | Maximum value, standard deviation, mean value and difference between the maximum and minimum values of LST | LST_Max, LST_Std, LST_Avg, LST_Range |

*3.2. Multi-Source Data Classification Based on FL-LightGBM*

LightGBM, a distributed gradient boosting framework based on the decision tree algorithm proposed by Microsoft in 2017, which is primarily based on the gradient boosting decision tree (GBDT) algorithm, uses two algorithms that improve performance: gradient-based one-side sampling (GOSS) and exclusive feature bundling, which is a more efficient implementation of GBDT [42]. LightGBM is often used in machine learning and data mining tasks [28,43] because it yields reliable results regarding classification, prediction, and ranking with high accuracy and efficiency [42].

The FL function was proposed [44] to address the challenge of sample imbalance in target detection. Additionally, it can be used at a later stage as a loss function in classification algorithms to adjust the weights of difficult samples during training. This loss function improves the classification performance of the algorithm by reducing the weight of the majority samples and increasing the weight of the minority samples during training, based on the standard cross-entropy loss function.

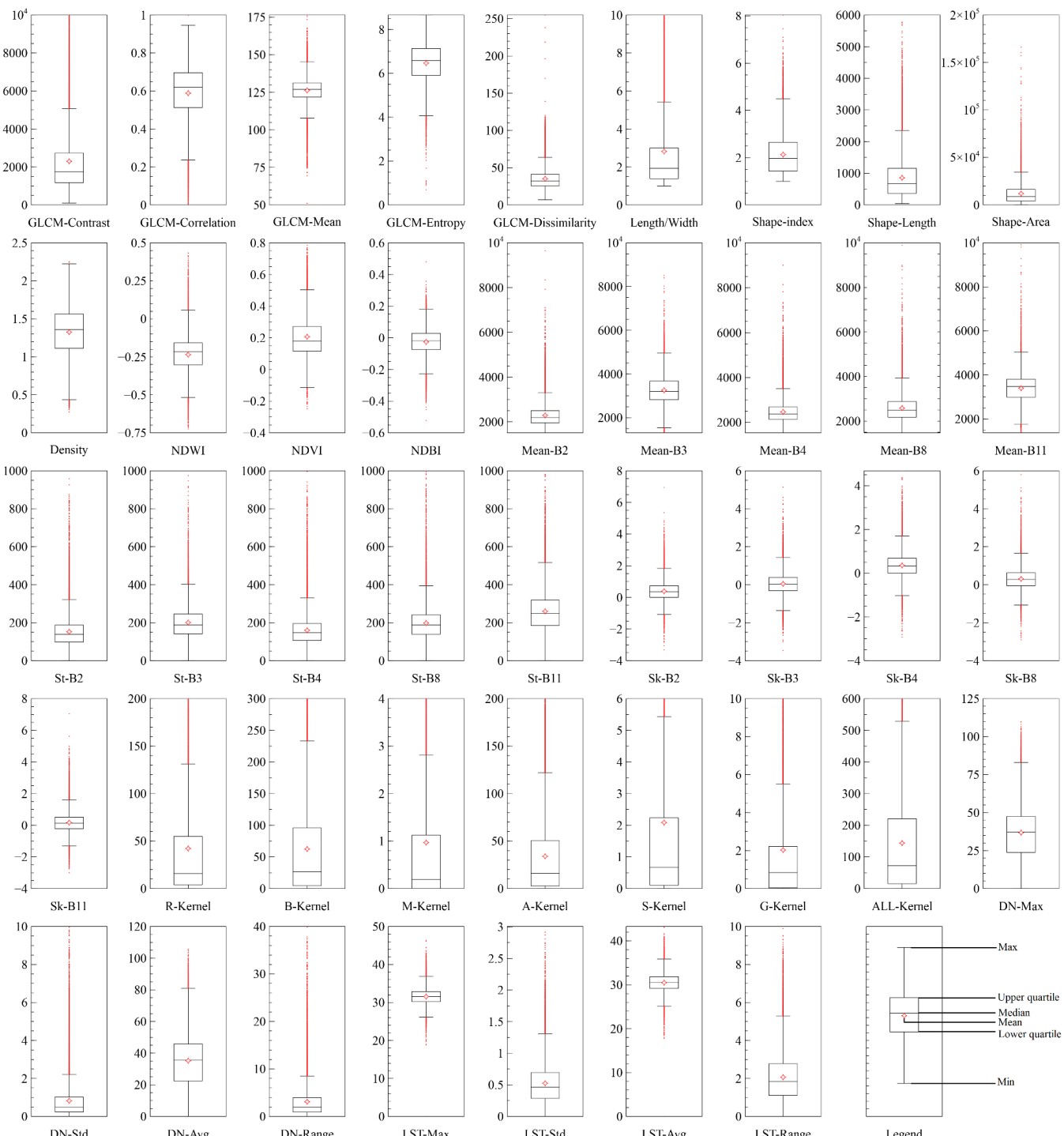

**Figure 3.** Box plot distribution of the 43 extracted data features (i.e., indicating the characteristics of the distribution of values for each data feature within a certain range).

The FL-LightGBM algorithm replaces the default cross-entropy loss function in the LightGBM algorithm with the FL function, enabling the LightGBM algorithm to place additional focus on minority class samples and indistinguishable samples by adjusting the category weighting factor $\alpha$ and the difficulty weighting factor $\gamma$. Here, FL was applied to the multivariate classifier of LightGBM, and a binary classifier was trained for each category C. Data from category C were treated as 1, and all other data were treated as 0. This method improves the sample category imbalance problem and increases the accuracy

of the classification model. The expression for the focal loss function after the introduction of the category weighting factor $\alpha$ and the difficulty weighting factor $\gamma$ is as follows:

$$P_i = \begin{cases} -\alpha(1-y')^{\gamma}\ln y', & y = 1 \\ -(1-\alpha)y'^{\gamma}\ln(1-y'), & y = 0 \end{cases} \tag{5}$$

where $\alpha$ is the category weighting factor, $\alpha \in (0,1)$, $\gamma$ is the difficulty weighting factor, y denotes the label of the true sample, and y' denotes the predicted value.

### 3.3. Division of Basic Research Units Based on OSM Road Network Data

Roads form the skeleton of a city, the backbone of all types of buildings and activity spaces, and they play a guiding role in urban development. Heiden and Yu et al. [45,46] were the first to propose the use of road network data to extract functional areas, as the blocks divided by road networks form the basic units of urban morphological structure, cognition, and management of urban functions, and they are widely used in urban analysis with clear functional implications [47,48].

Road network data include many types of roads. Thus, when using road network data to segment a study area, selecting an appropriate road category is necessary to construct the road network. In this study, we utilized the block division method based on OSM road network data [49] and combined it with the actual screening of roads to establish rules and to delineate the basic study units. The specific method was as follows: (1) road information was filtered, and redundant roads were removed; (2) the road centerline was extracted using ArcGIS, and certain road areas were grouped into blocks; (3) road network topology inspection and error modification were performed to create a complete and closed road network; and (4) the road network was refined and blocks were constructed to better define the edges of blocks in conjunction with elements such as rivers and other water bodies.

### 3.4. Determining UFZs Types Based on Block Units

The multimodal data classification results based on the FL-LightGBM were reconstructed at the block scale to determine the nature of the UFZs by constructing a category ratio (CR). The formula is as follows:

$$R_i = \frac{A_i}{\sum_{i=1}^{7} A_i} \times 100\% \tag{6}$$

where $R_i$ represents the proportion of the area of type $i$ classification results to the area of all types in the cell, and $A_i$ is the area of type $i$ classification results in the cell.

When $R_i \geq \alpha$, the unit was determined to be a single functional area of type $i$. Based on empirical and experimental analysis, $\alpha$ was set at 0.4 for commercial land (B) and transport land (S), 0.5 for residential land (R), and 0.6 for industrial land (M), administrative and public services land (A), scenic areas and squares land (G), and other types of land (E). Otherwise, the unit was determined to be a mixed functional area, with the type of mix depending on the two dominant functional types within the unit.

### 3.5. Block-Scale Accuracy Verification

In this study, the accuracy of the UFZ identification results at the block scale was verified using an error matrix that included four main accuracy metrics: user accuracy, identification accuracy, overall accuracy, and the kappa coefficient. The formulae are as follows:

$$\text{User accuracy} = X_{ii}/X_{i+} \tag{7}$$

$$\text{Recognition accuracy} = X_{ii}/X_{+i} \tag{8}$$

$$\text{Overall accuracy} = \sum_{i=1}^{r} X_{ii}/M \times 100\% \tag{9}$$

$$K = (M\sum_{i=1}^{r} X_{ii} - \sum_{i=1}^{r} X_{i+}X_{+i})/(M^2 - \sum_{i=1}^{r} X_{i+}X_{+i}) \tag{10}$$

where $K$ is the kappa coefficient; $r$ is the number of rows in the error matrix; $X_{ii}$ is the value of row $i$ and column $i$; $X_{i+}$ and $X_{+i}$ are the sums of rows $i$ and $i$, respectively; and $M$ is the total number of samples tested.

### 3.6. Methods of Spatial Structure Analysis

3.6.1. Location Entropy

Location entropy is an important indicator of the spatial distribution of factors in a region and the degree of specialization in a particular sector [50]. In the study of the spatial distribution of urban functions, the location entropy index was primarily used to analyze the agglomeration of dominant functions in the region. The calculation function was as follows:

$$LQ_{ij} = \frac{S_{ij}/S_j}{S_i/S} \tag{11}$$

where $LQ_{ij}$ denotes the location entropy of element $j$ of unit $i$ within the study area, $S_{ij}$ denotes the area of functional element $j$ in unit $i$, $S_j$ denotes the area of functional element $j$ in all units, $S_i$ denotes the sum of the areas of all functional elements in unit $i$, and $S$ denotes the sum of the areas of all functional elements in all units. When $LQ_{ij} > 1.0$, element $j$ of unit $i$ has the advantage of agglomeration within the study area and forms a functional area. Alternatively, when $LQ_{ij} > 1.5$, element $j$ of unit $i$ has the advantage of significant agglomeration within the study area and forms a specialized functional area.

3.6.2. Compound Degree

Based on the results of the location entropy calculation, the location entropy values of the seven functional elements (R, B, M, A, S, G, and E) were reclassified and assigned (if $LQ > 1$, value (compound degree) = 1; if $LQ < 1$, value (compound degree) = 0). A superposition operation was performed to obtain the compound degree of the urban functions of the block to determine the regional aggregation advantage of the functional elements within the block.

## 4. Experiments and Analysis

### 4.1. Recognition Performance with Different Model Default Parameters

Six common classification models were selected: 10% of the data in the study area was extracted for labelling to create the dataset, and the training and test sets were trained and tested in a 7:3 ratio to compare their model accuracy and time spent with a 5-fold cross-validation (other parameters were set at default). For the experimental environment, Windows 10 (processor i5-8250U, 8 Gb RAM), Python 3.8, and Jupyter Notebook platforms were used. The results are shown in Figure 4, which indicates that LightGBM is highly efficient and accurate.

### 4.2. FL-LightGBM Based Classification Experiments

4.2.1. Prediction Accuracy of Various UFZ Types

In this study, we used Optuna [51] to tune hyperparameters to optimize Light-GBM, and the corresponding main model parameters 'n_estimators', 'learning_rate', 'num_leaves', 'feature_fraction', and 'max_depth' were 2342, 0.047, 79, 0.586, and 8, respectively. Additionally, we simultaneously finetuned $\alpha$ and $\gamma$ to obtain a robust FL-LightGBM. To evaluate the impact of $\alpha$ and $\gamma$ values on the algorithm results, $\alpha$ was set to [0, 10] and $\gamma$ to [0, 5]. Different combinations of parameters were used to evaluate the prediction accuracy of the final results. The results are shown in Figure 5a. The highest accuracy was achieved when the values of ($\alpha$, $\gamma$) were (0.15, 5), indicating that the parameter values at this point have improved classification results for the FL-LightGBM algorithm.

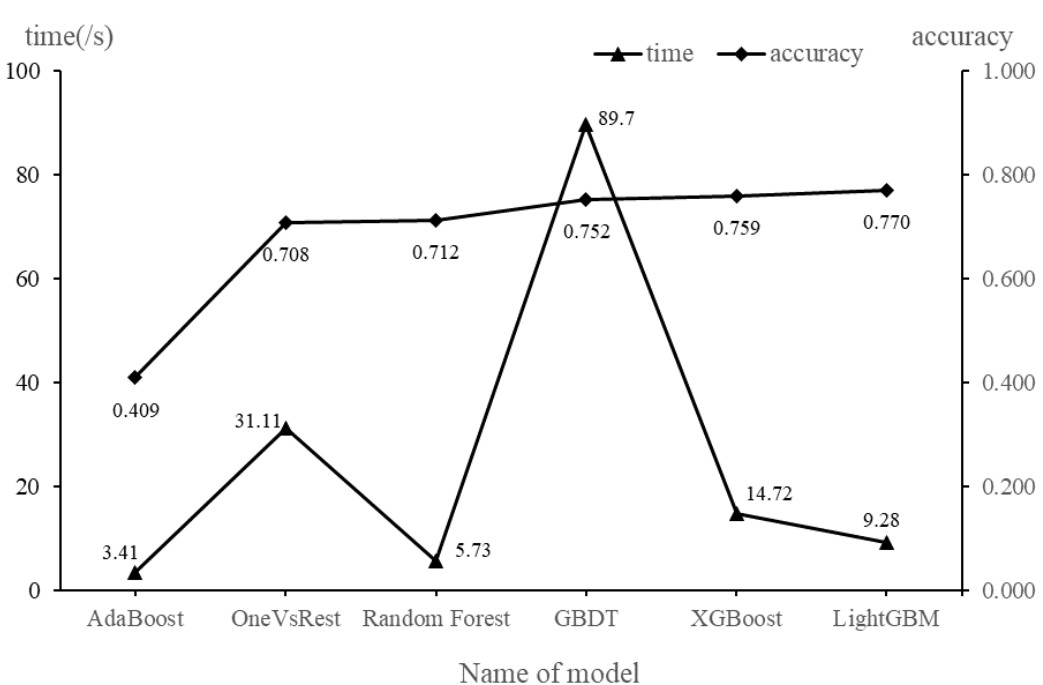

**Figure 4.** Comparison of the performance of different classification models.

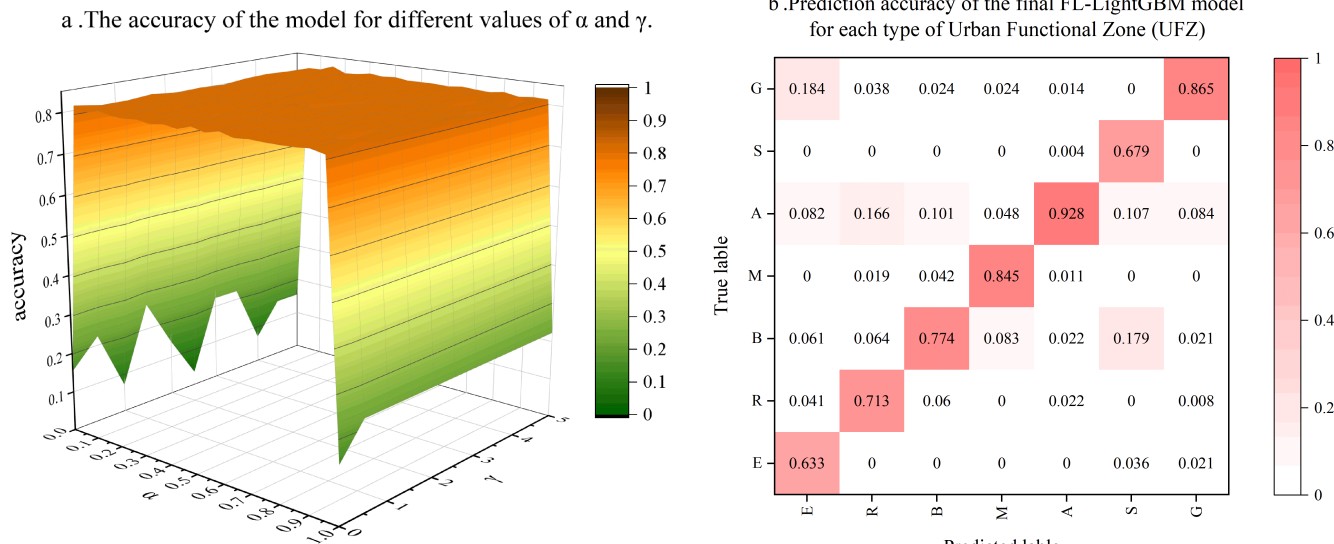

**Figure 5.** (**a**) Accuracy of the model for different values of α and γ, and (**b**) Prediction accuracy of the FL-LightGBM mode for each type of Urban Functional Zone (UFZ).

Figure 5b shows the classification accuracy for various functional area types using the FL-LightGBM algorithm, with an overall accuracy of 0.8253 and a recall of 0.7767. The values for the individual grids were significantly higher on the diagonal than on the non-diagonal grid, indicating that a much higher proportion of UFZs were correctly classified than incorrectly classified. The values of M, A, and G were >0.8, indicating the highest classification accuracy. In the non-diagonal section, E was misclassified as G, R as A, and S as B at a high rate.

4.2.2. Comparison Test

Figure 6 shows a comparison of the overall classification accuracy and recall for different combinations of data sources; the classification accuracy was lower for single-source data when compared with combination data. Among the four types of data, the

classification accuracy of POI was significantly higher than that of Sentinel-2, nighttime light, and LST, and higher than that of any combination of the other three types of data. Compared with the POI single-source data, the incorporation of Sentinel-2, nighttime light, and LST data improved the recognition accuracy by 3.29%, 3.29%, and 3.39%, respectively. The highest classification accuracy was achieved by fusing the four types of data, which resulted in increases of 27.8%, 10.1%, 48.8%, and 48.5% compared with those achieved using Sentinel-2, POI, nighttime light, and LST single-source data, respectively.

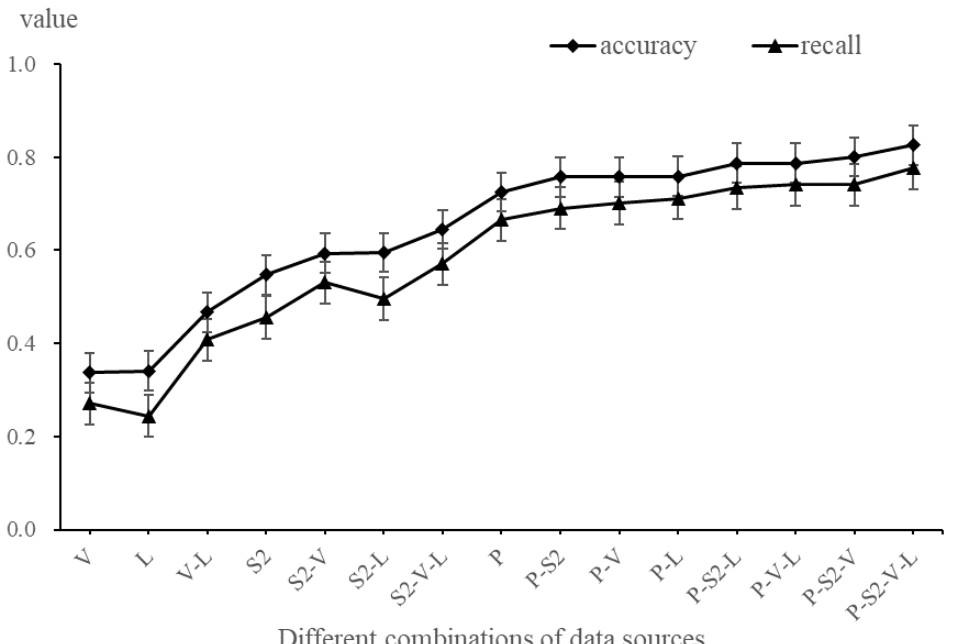

**Figure 6.** Comparison of classification accuracy for different combinations of data sources. S2, P, V, and L in the figure indicate Sentinel-2, point of interest (POI), NPP-VIIRS nighttime light, and land surface temperature (LST) data sources, respectively.

The feature importance analysis under the combination of the four data sources (Figure 7) shows that each data feature derived from the POI data has a high importance score, indicating that the features that combine the semantic information of the three POI levels can accurately express the characteristics of the UFZs. This analysis was followed by data features derived from the LST (at the seventh and eleventh positions) and nighttime light data (at the ninth and tenth positions). The importance scores of individual data features derived from Sentinel-2 data (after the eleventh position) were lower than the importance scores of individual data features derived from the first two positions, indicating that individual data features derived from LST and nighttime light data had a higher sensitivity in urban functional zoning than individual data features derived from Sentinel-2. However, in terms of classification accuracy, after combining the four data types, the improvement over Sentinel-2 data was higher than that over POI data. One possible reason for this is the limited spatial resolution of the Sentinel-2 experiment in this study, such that only a small amount of classification-identification information could be extracted. The classification accuracy and feature importance scores after fusing nighttime light and LST data showed that these data types could provide a unique perspective for identifying urban functional areas.

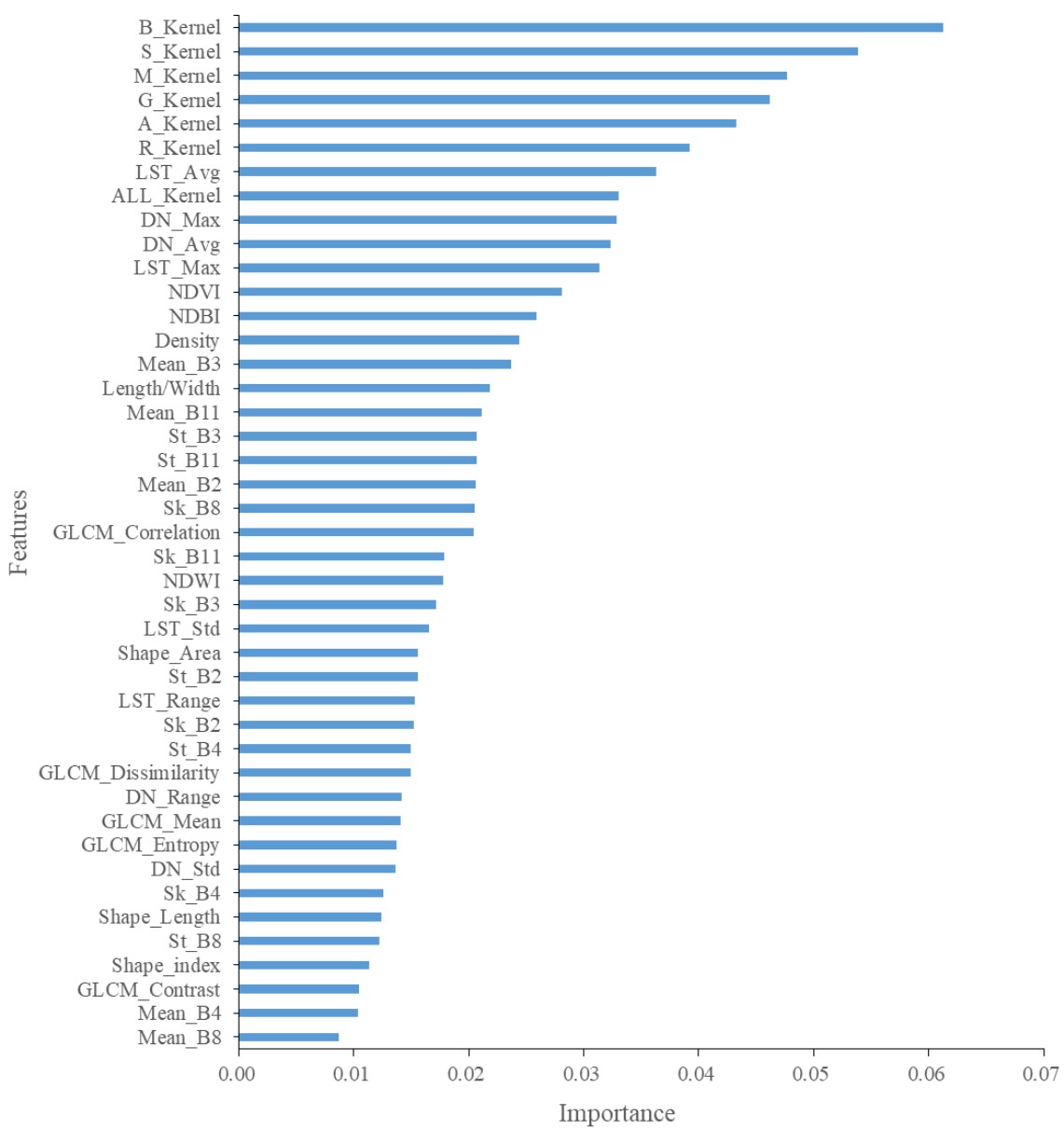

**Figure 7.** Ranking of feature importance for four combinations of data sources.

### 4.3. Classification Results Based on Block Reconfiguration

#### 4.3.1. Classification Results

The multiscale object classification results based on the FL-LightGBM classification model were reconstructed at the block scale to obtain the functional zoning results. First, we performed a secondary partitioning of the object units using the block plot units obtained in Section 3.3 to ensure that an object was within a block unit. We then determined the type of urban functional area at the block scale by counting the proportion of different urban functional areas within the block unit (i.e., the method described in Section 3.4) based on the classification of the object. Road areas were visualized by assigning widths of 25, 20, 15, 10, and 5 m to motorway, primary, secondary, tertiary, and unclassified roads, respectively, using OSM road network data. Rivers and waterways were also derived from the OSM data. As shown in Figure 8, the spatial distribution of the different UFZs is evident, with seven categories of single UFZs and one category of mixed UFZ identified (subdivided into 19 categories of mixed UFZs in different combinations, as shown in Figure 9b). The single UFZ was dominated by Residential land (R) (691 blocks in total, with a total area of 87.95 km$^2$, accounting for 15.57%) and Administration and public services land (A)

(544 blocks in total, with a total area of 92.47 km$^2$, accounting for 16.37%). A total of 452 blocks of Commercial land (B) (with a total area of 49.31 km$^2$, accounting for 8.73%) were identified, whereas few plots of Scenic areas and squares land (G) and Other land (E) were identified but had a larger area, with a total area of 57.53 km$^2$ and 39.17 km$^2$, respectively. Industrial land (M) had 72 blocks, with a total area of 10.57 km$^2$, whereas Transport land (S) had 42 blocks, with a total area of 4.02 km$^2$. Mixed UFZ accounted for the largest overall proportion, with 676 blocks and a total area of 223.79 km$^2$, accounting for 39.62%.

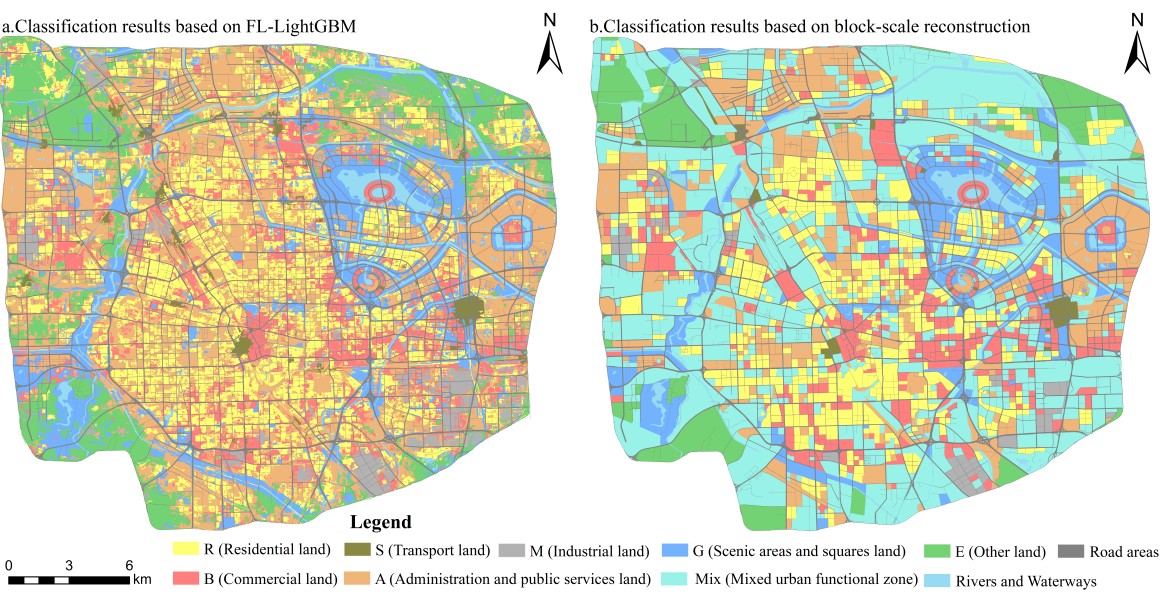

**Figure 8.** Graph of classification results. Mix indicates mixed urban functional zone.

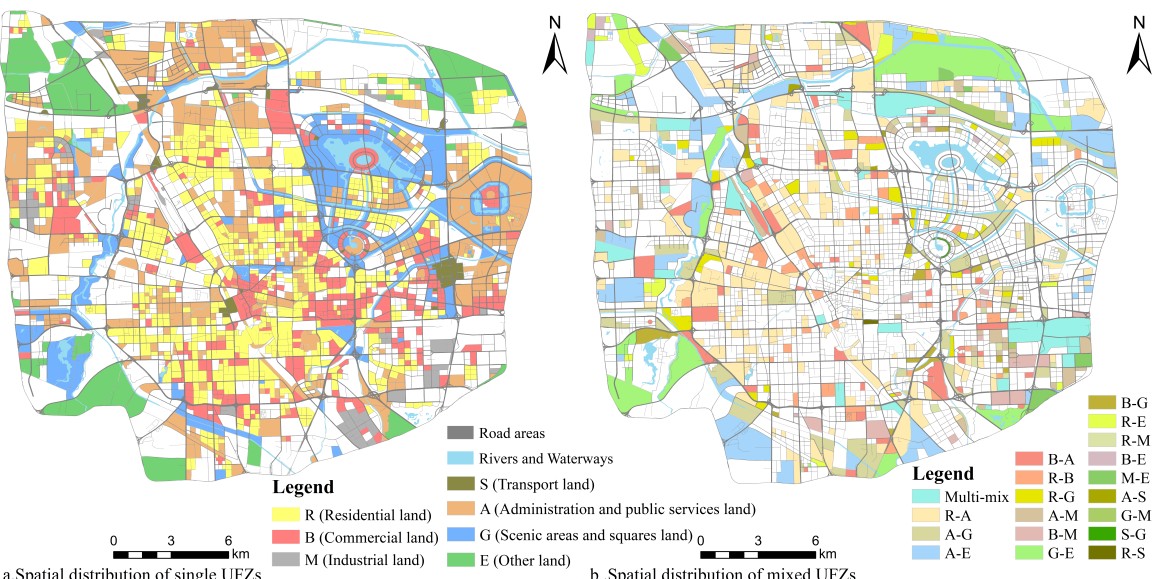

**Figure 9.** Distribution of urban functional zones (UFZs). Multi-mix indicates multi-mixed UFZ.

### 4.3.2. Block-Scale Accuracy Verification

To verify the accuracy of the UFZ identification results at the block scale, 225 blocks were randomly selected and compared with AutoNavi Maps to determine the true attributes of the UFZs. The accuracy of the results was verified using an error matrix to produce accurate classification results (Table 3).

**Table 3.** Error matrix for the validation of urban functional zones (UFZs).

| Identified \ Actual | Residential Land | Commercial Land | Industrial Land | Administration and Public Services Land | Transport Land | Scenic Areas and Squares Land | Other Land | Mix | Total | Recognition Accuracy |
|---|---|---|---|---|---|---|---|---|---|---|
| Residential land | 26 | 1 | 0 | 2 | 0 | 0 | 0 | 1 | 30 | 0.8667 |
| Commercial land | 2 | 25 | 1 | 1 | 0 | 0 | 0 | 1 | 30 | 0.8333 |
| Industrial land | 1 | 2 | 25 | 1 | 0 | 0 | 0 | 1 | 30 | 0.8333 |
| Administration and public services land | 1 | 1 | 1 | 25 | 0 | 1 | 0 | 1 | 30 | 0.8333 |
| Transport land | 0 | 1 | 0 | 1 | 12 | 0 | 1 | 0 | 15 | 0.8000 |
| Scenic areas and squares land | 1 | 1 | 0 | 0 | 0 | 26 | 0 | 2 | 30 | 0.8667 |
| Other land | 0 | 0 | 0 | 1 | 0 | 2 | 27 | 0 | 30 | 0.9000 |
| Mix | 1 | 1 | 1 | 1 | 0 | 2 | 1 | 23 | 30 | 0.7667 |
| Total | 32 | 32 | 28 | 32 | 12 | 31 | 29 | 29 | 225 | |
| User accuracy | 0.8125 | 0.7813 | 0.8929 | 0.7813 | 1.000 | 0.8387 | 0.9310 | 0.7931 | | |
| Overall accuracy = 84.00% kappa = 0.8162 | | | | | | | | | | |

Table 3 shows that the overall accuracy was 85.19% with a kappa coefficient of 0.8027, indicating that the model identification results were highly consistent with the actual situation, and they are therefore reliable.

### 4.4. Analysis of the Layout of UFZs in Zhengzhou

#### 4.4.1. Spatial Distribution Characteristics of Single UFZs

A single UFZ was dominated by A (administration and public services land) and R (residential land) (Figure 9a). A (administration and public services land) was mostly distributed in areas with a high concentration of universities and showed a predominance of government units and hospitals. In contrast, R (residential land) was mostly distributed in the peripheral areas of the center and generally showed the layout characteristics of a fan-shaped structure spreading outwards. B (commercial land) was characterized by a "polycentric" cluster in the Erqi Square commercial area, the intersection of Longhai Road, Zhongzhou Avenue, and Longhu Centre, as it is easy to create a commercial agglomeration effect in areas of busy economic activity. M (industrial land) and S (transport land) were small in area and single in distribution. M (industrial land) was mainly distributed in the peripheral areas of the city, while S (transport land) mainly presented a double-center structure dominated by Zhengzhou Railway Station and Zhengzhou East Railway Station. G (scenic areas and squares land) and E (other land) were mainly distributed on the periphery of the city and had a large average land area.

#### 4.4.2. Spatial Distribution Characteristics of Mixed UFZs

The diversity of UFZs comprise a key component of urban complexity [52], with diversity reflecting heterogeneous economic and socio-spatial structures [53]. As shown in Figure 9b, mixed UFZs were predominantly prevalent within the study area, and the complexity of the UFZ mix increased as the city continued to expand in size. Among the mixed UFZs, R-A (residential land-administration and public services land) predominated, followed by B-A (commercial land-administration and public services land) and R-B (residential land-commercial land). Residential and commercial functions are inextricably linked, as shown in Figure 9a, where the phenomenon of "residential-commercial linkages" is prevalent. Residential land (R), Commercial land (B), and Administration and public services land (A) form a linkage, indicating that the commercial and service configuration of the residential community is currently increasing. By forming a linkage with commercial and public services, the daily needs of residents can be met, forming a complex characterized by the linkage of urban residential with commercial and public services. The remaining mixed lands were extremely discrete in distribution and fewer in number.

#### 4.4.3. Analysis of the Compound Features of the Functional Space

The compounding of urban functions involves spatial aggregation of multiple interacting functional elements. This study uses the block as the unit scale and introduces

location entropy, equilibrium, and compound degree indices to further study the compound characteristics of functional areas within the urban unit.

Figure 10a–g shows the spatial distribution characteristics of the locational entropy of different functional elements, with a relatively high number of units having significant advantages in the R (residential land), B (commercial land), and A (administration and public services land) functional elements, with 1275, 1041, and 1152 block units, respectively. These values were followed by 624 units with significant advantages in the G (scenic areas and squares land) category, while the M (industrial land) and E (other land) categories accounted for a relatively small number of units and were mostly located in the periphery of the city, with the lowest number of units (n = 191) retrieved from the S (transport land) category.

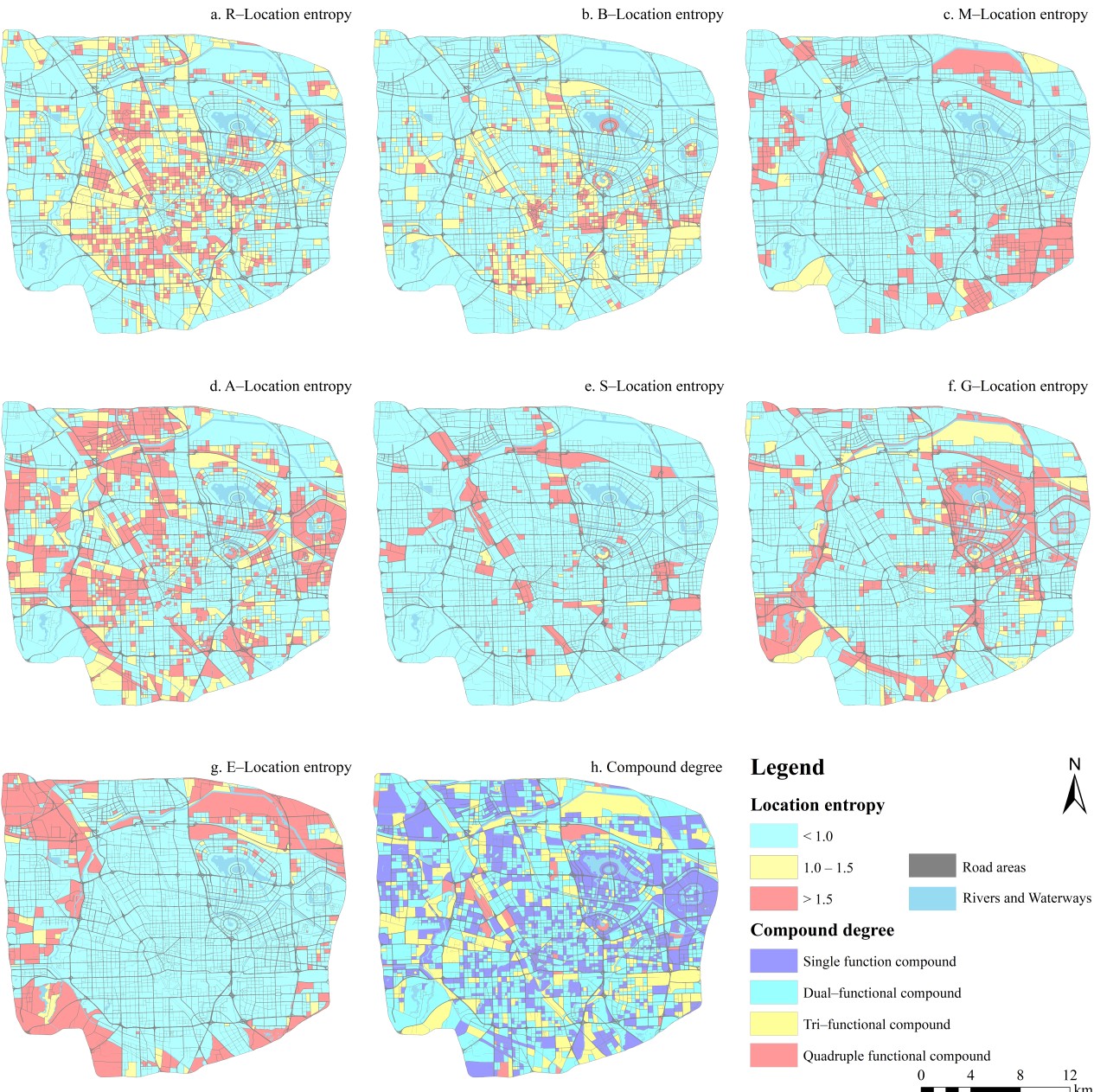

**Figure 10.** Calculated location entropy (**a–g**) and compound degree (**h**) of each functional element in the study area.

The geographical distribution of the different functional elements was further analyzed by calculating the average locational entropy values of each functional element within the different loop road areas, as shown in Table 4. The most advantageous type of functional element R (residential land) was found in the second loop road area. This finding was owing to the Erqi Square commercial area and Zhengzhou Station both being located in the first-loop road area. Subsequently, the B (commercial land) and S (transport land) functions in the first-loop road area are situated more advantageously, as the residential areas are distributed outward from the city center. The advantages of A (administration and public services land) are relatively even, indicating that the people's livelihood infrastructure and service organization system in the city are relatively complete, and that there is a high level of scientific, educational, and cultural facility construction. The dominant areas of M (industrial land) are clustered in the fourth loop road area, and the dominance of G (scenic areas and squares land) was distributed between the third and fourth loop road areas. E (other land) was lower overall, with the highly dominant areas being distributed along the fourth loop road area.

**Table 4.** Average locational entropy values for each functional element within each loop road area.

| Regional Scope | Number of Blocks | R (Residential Land) | B (Commercial Land) | M (Industrial Land) | A (Administration and Public Services Land) | S (Transport Land) | G (Scenic Areas and Squares Land) | E (Other Land) |
|---|---|---|---|---|---|---|---|---|
| First loop | 149 | 1.091 | 3.638 | 0.000 | 0.720 | 2.372 | 0.381 | 0.000 |
| Second loop | 454 | 1.949 | 1.908 | 0.203 | 0.939 | 0.245 | 0.231 | 0.000 |
| Third loop | 896 | 1.434 | 1.648 | 0.789 | 0.848 | 1.682 | 1.006 | 0.014 |
| Fourth loop | 1297 | 0.984 | 0.915 | 1.342 | 1.151 | 1.228 | 0.821 | 0.813 |

The results of the zone locational entropy calculation were used to further analyze the level of functional element compounding within the block units. As shown in Figure 10h, single- and dual-functional compounds had the highest proportions of composites. We thus counted the number of blocks with locational entropy values greater than one for a single type of functional element and those with locational entropy values greater than one for two or more functional elements. The results (Figure 11) showed that the number of blocks with a single-functional element compound and a dual-functional element compound was 1142 and 1283, respectively, leaving a total of 371 blocks with multiple-functional element compounds. This result indicated that the coordinated development of single- and dual-functional compounds was the main form of block-scale expression. Statistical analyses showed that the majority of single-functional compounds are dominated by R (residential land), B (commercial land), and A (administration and public services land) blocks, followed by G (scenic areas and squares land), and relatively few by M (industrial land), S (transport land), and E (other land) blocks. Residential land administration and public services land (R-A) and residential and commercial land (R-B) compounds predominated among the dual-functional compounds, indicating strong spatial compatibility of the R (residential land), B (commercial land), and A (administration and public services land) functions, resulting in a strong degree of R (residential land), B (commercial land), and A (administration and public services land) aggregation in most areas.

Overall, the development levels of various functional elements at the block scale varied considerably. One block scale had distinct advantages for one functional element, and the advantageous areas of each function showed a diversified and fragmented distribution pattern at the block scale, indicating that the UFZ at the block microscale had a significant diversification attribute.

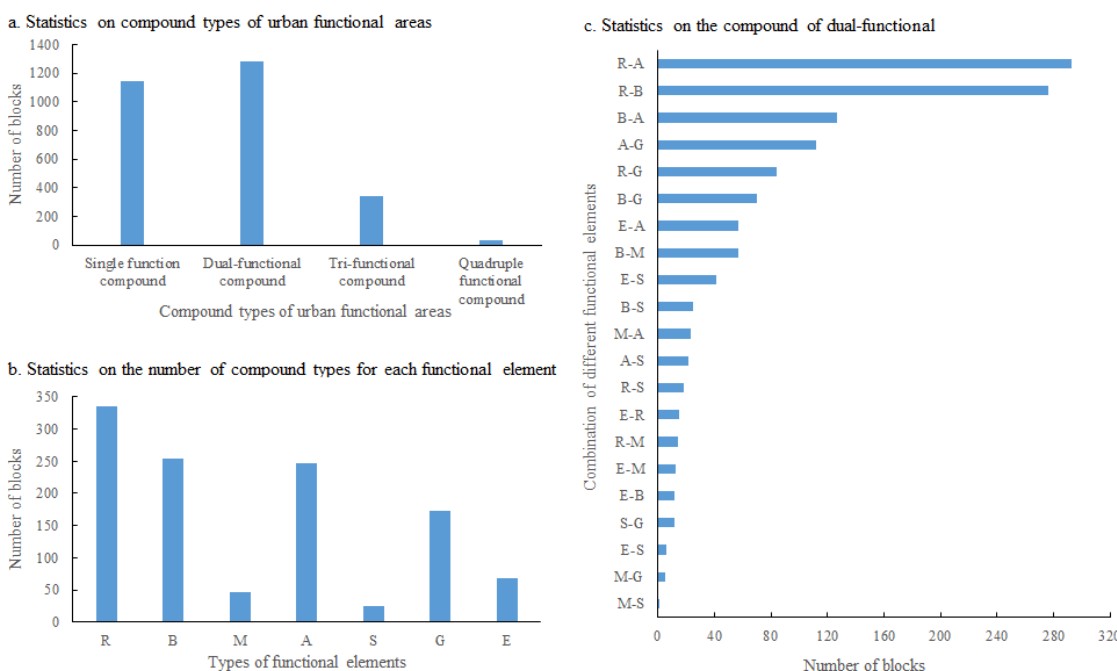

**Figure 11.** Statistics on compound types. (**a**) Compound types of urban functional areas; (**b**) Number of compound types for each functional element; and (**c**) Compound of dual-functional.

## 5. Discussion

Data such as remote sensing images [54] and mobile phone signaling [55,56] perform well in identifying urban functional areas; however, their description of urban functional areas is not comprehensive, whereas POI data comprise the limitation of spatial sparsity. This study fully considered the characteristics of each data feature and identified urban functional areas by fusing the feature information from different data sources to analyze the urban functional layout of the central city of Zhengzhou. First, in terms of data feature fusion, the fusion of multiple sources of data can significantly improve the accuracy of functional area recognition, with a maximum improvement of 48.8% when compared with single-sourcedata (e.g., nighttime lights). The highest accuracy of 72.46% was achieved using only single-source POI data compared with other single-source data, and the POI data features performed the best in the ranking of importance of the experimental data features, indicating that the POI three-level semantic index constructed in this study can accurately characterize urban functional areas to a greater extent. This method addresses the shortcomings of POI data in identifying urban functional areas without geographical entity areas or influence ranges. However, when reconstructing the results at the block scale, the proportion of individual functional areas influences the functional attributes of the block. For example, most areas had a higher density of commercial facility points, and their classification results were more likely to be classified as commercial functions (however, the true attributes should comprise residential or other functions). Therefore, we fine-tuned the setting of 50% as the threshold to improve this result (Section 3.4); however, we did not perform a highly specific and detailed analysis, which should be accounted for in future studies. Second, in terms of the selection of research scale units, this study reconstructed multi-scale segmented patches at the block scale to complete the functional area identification of the final block units with an accuracy of 84% in the final results. Compared with using regular grid division [27,57] for functional zoning experiments, the latter reconstruction not only represented the most basic unit scale of a city, but also better expressed the functional nature of a unit parcel [25]. Using multilevel roads to construct blocks as the basic unit of urban functional area division, this approach avoids grid size determination and the impact of road segmentation, thereby effectively preserving the integrity of the urban functional area at the block scale [55,58]. Finally, we proposed a new

framework for the classification of urban functional areas based on multimodal data and the FL-LightGBM algorithm. This proposal is based on the fusion of multi-source data characteristics, considering the imbalance of samples, and it is capable of the rapid and accurate identification of urban functional areas. In Zhengzhou City, the accuracy of the experimental results of the data fusion method in this study was improved compared with other methods prevalent in the literature [28,59]. Furthermore, the identification results of the urban functional areas closely resembled that of reality and it has high practical value for the rapid recognition of modern urban functional areas.

This study identified not only single-functional areas but also dual-functional mixed and multifunctional mixed-use blocks, whose mixed functional areas are a key context for urban complexity and a concrete manifestation of socio-economic-environmental interactions [52]. Through the analysis of the urban spatial structure, understanding the differentiation and locational advantages of urban functional areas helps city planners adjust their policies in a timely manner, account for the advantages of regional location and renewable resources, promote reasonable flow and efficient concentration of various factors, and form a regional economic layout of diversified industries, which is of great significance in integrating effective urban resources and promoting balanced and sustainable urban development.

Although the method of fusing multiple sources of data features in this study achieved the expected results and provided a basis for the rapid and accurate identification of urban functional areas, certain limitations remain. First, the spatial resolution of the Sentinel-2 multi-spectral remote sensing imagery used in this study was 10 m, which caused bias in the extraction of target feature patches. In the future, very-high-resolution (VHR) images can be used to segment and extract fine feature patches in combination with light detection and ranging (LiDAR) data to extract three-dimensional urban information to obtain large amounts of feature information for urban functional zone recognition. Second, a certain deviation was present in the use of POI and nighttime light imagery to characterize socioeconomic characteristics in urban functional zones because POI majorly exists in areas with high building density, and nighttime light imagery can only record the economic activities of the human society at night. The nighttime light imagery used in this study had a low spatial resolution, for which higher resolutions could be used in the future to improve the method's contribution to urban functional zone recognition. In addition, according to relevant studies [28,56], time information features also make important contributions to the identification of urban functional zones, whereas the present study does not use data with time-series characteristics (e.g., Time-Series User Behavior Data and mobile phone signaling data). Therefore, future research should consider the temporal characteristics of urban functional zones to fully explore the temporal and spatial characteristics of urban functional zones.

## 6. Conclusions

This study proposes a method for classifying UFZs based on the FL-LightGBM algorithm fusing multi-source data to spatially identify the four-ring area of Zhengzhou and to analyze the distribution characteristics of different UFZs within the city.

The main findings are summarized as follows.

(1) This study used FL-LightGBM to fuse multi-source data features for model training and prediction based on the multi-scale segmentation of remote sensing images with an accuracy rate of 0.8253, which can effectively identify the types of UFZs at multiple scales. By examining the reconstruction results at the block scale, the classification accuracy of urban functional areas reached 84%, and the kappa coefficient reached 0.8162, indicating that the recognition results of the method were highly consistent with the actual situation, and are therefore feasible.

(2) The integrated semantic information of the three POI levels could better characterize the semantic information of UFZs. The incorporation of multi-spectral, nighttime

light, and LST data further improved the recognition accuracy by approximately 10.1% compared within single-source POI data.

(3) The overall layout of the main urban area of Zhengzhou showed the coordinated development of single and mixed UFZs, in which a more distinct R-B-A complex feature was formed, and the UFZs at the microscopic scale of the block had diverse attributes.

The identification of UFZs is fundamental for understanding spatial patterns within a city. In response to the problems of POI data being weak in identifying UFZs in areas with low building density and sparse data, remote sensing data lacking the necessary semantic information for functional zoning, and single-source data being unsuitable for performing a more comprehensive characterization of complex UFZs, this study combined remote sensing images, POI, nighttime light, and LST multi-source data for functional partitioning and achieved improved results. By combining multiple features from multiple sources of heterogeneous data, highly accurate and comprehensive characterizations of complex urban functional areas can be achieved, which is important for the rapid and accurate identification of urban functional areas. This result aids in mapping urban land cover and functional areas and is important for urban surveys and management. Although the complementary advantages of various types of geographical data can improve the accuracy of identifying functional urban areas, cities are highly heterogeneous and complex systems, and the functionality of their blocks may change over time. The issue of deeper mining of the dynamics of structural functions within the city and the interaction of space and function needs to be explored further in conjunction with other multi-source sensory data (e.g., mobile phone signaling data and taxi tracking data).

**Author Contributions:** Conceptualization, C.G.; methodology, C.G. and J.W.; validation, C.G., M.W. and Y.Z.; formal analysis, C.G., Y.Z. and M.W.; investigation, C.G.; resources, M.W. and Y.Z.; data curation, C.G.; writing—original draft preparation, C.G. and J.W.; writing—review and editing, C.G. and J.W.; visualization, C.G.; supervision, J.W.; All authors have read and agreed to the published version of the manuscript.

**Funding:** This study received no external funding.

**Institutional Review Board Statement:** Not applicable.

**Informed Consent Statement:** Not applicable.

**Data Availability Statement:** All Data can be found in Section 2.2.

**Conflicts of Interest:** The authors declare no conflict of interest.

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
