# Peer review of "Identification of Urban Functional Areas and Urban Spatial Structure Analysis by Fusing Multi-Source Data Features: A Case Study of Zhengzhou, China"

_sustainability, doi:10.3390/su15086505_

Round 1

Reviewer 1 Report

The topic and research problem that addressed in this manuscript is important and relevant to the research in sustainable urban planning. The proposed data and methodology can be an attribute to the traditional use of remote sensing. However, the manuscript needs some improvement. The following are my comments that can be addressed.

1.     In the introduction section, the author describes the research gap based on existing studies and the referenced methods. However, the importance of combining multiple sources of data is not clearly explained.

a)     How is it beneficial and why should be considered if the existing methods including the classification of high resolution of remote sensing images can provide the accurate urban functional areas?

b)    Author should elaborate more on why The Focal Loss LightGBM algorithm was selected to perform the classification.

2.     Section 2, the subtitles under 2.2 can be restructured. Either use third subtitles such as 2.2.1 or another better format to facilitate the readers.

3.     Section 2.3 this title of “Research ideas” is confusing. Find an appropriate terminology and can be merged with methodology section 3.

4.     Figure 3. Author need to provide a short description in the caption to guide the readers to what the figure illustrates.

5.     Figure 11 is not readable. Need to be improved.

Author Response

Dear Reviewer,

We wish to thank you for your insightful comments. These have greatly helped us to improve the quality of our manuscript.

In accordance with your comments, we have made the following main changes:

Point 1: In the introduction section, the author describes the research gap based on existing studies and the referenced methods. However, the importance of combining multiple sources of data is not clearly explained. a) How is it beneficial and why should be considered if the existing methods including the classification of high resolution of remote sensing images can provide the accurate urban functional areas? b) Author should elaborate more on why The Focal Loss LightGBM algorithm was selected to perform the classification.

Response 1: It is expanded in the introduction section. The importance of combining multiple data sources is illustrated(lines 59-64 and lines 97-103) and the reason why Focal Loss LightGBM algorithm was chosen for the classification is explained(lines 107-125).

Point 2: Section 2, the subtitles under 2.2 can be restructured. Either use third subtitles such as 2.2.1 or another better format to facilitate the readers.

Response 2: Modified the subtitles under sections 2.2 and 3.1 to the third subtitle. such as 2.2.1 and 3.1.1.

Point 3: Section 2.3 this title of “Research ideas” is confusing. Find an appropriate terminology and can be merged with methodology section 3.

Response 3: Section 2.3 has been deleted and merged into the first paragraph under Section 3 Methodology.

Point 4: Figure 3. Author need to provide a short description in the caption to guide the readers to what the figure illustrates.

Response 4: Modified the title of Figure 3 and added a short description.

Point 5: Figure 11 is not readable. Need to be improved.

Response 5: Modified the Figure 11: The original Figure 11.a was split into new figures 11.a and 11.b, and the original figure 11.b was modified into new figure 11.c.

Reviewer 2 Report

Review Report

sustainability-2281732

General notes:

 The study aimed to propose a method for identifying UFZs by fusing multi-attribute features from multi-source data and introduced nighttime light and LST indicators as functional zoning references in the urban area of Zhengzhou in China. The goal is interesting as the UFZs are very important in forming the urban fabric. The implementation of the proposed methodology is well presented and well implemented. The following are some comments to be considered.

Specific Comments:

Line 48-49: give examples about those certain factors.

Line 82-83: the statement needs to be rewritten to criticize the work rather than criticizing the authors.

Figure 1: the map of the study area is not informative. It is better to be replaced with real true color satellite image with the ring roads superimposed it for better comparison with the results.

Line 114: change the currency unit to US dollars for better comparison and understanding.

Paragraph Sentinel: what was the purpose of segmentation, what are the LULC types.

Line 134: more information about the NPP-VIIRS is needed e.g., spatial resolution. How it was incorporated in the analysis.

Section 3.1: add the equations of the spectral indices.

Figure 8: add the full name of the classes with the letters such as:  Residential land (R), and so on…

Table 3: add the full name of the classes in the columns and rows instead of letters.

Lines 390-402: should be moved to the Methodology section.

Section 4.4.1 and further sections: Please write the full name of the UFZ within the text instead of using the letters coding. Readers will not memorize what these letters mean, and they should not keep referring to Table 1.

Author Response

Dear Reviewer,

We wish to thank you for your insightful comments. These have greatly helped us to improve the quality of our manuscript.

In accordance with your comments, we have made the following main changes:

Point 1: Line 48-49: give examples about those certain factors.

Response 1: We give examples of those certain factors in lines 49-52.

Point 2: Line 82-83: the statement needs to be rewritten to criticize the work rather than criticizing the authors.

Response 2: Line 91-92: Rewrote the statement to state the shortcomings of the work.

Point 3: Figure 1: the map of the study area is not informative. It is better to be replaced with real true color satellite image with the ring roads superimposed it for better comparison with the results.

Response 3: We have modified Figure 1 by using a real true color satellite image for the base image instead.

Point 4: Line 114: change the currency unit to US dollars for better comparison and understanding.

Response 4: Line 148: Changed currency unit to USD.

Point 5: Paragraph Sentinel: what was the purpose of segmentation, what are the LULC types.

Response 5: The purpose of the multi-scale split is explained in lines 166-169. In addition we do not understand what is meant by "what are the LULC types" and we do not know whether you are asking about the types of urban land use. We have regrouped the functional urban land types into seven categories in Table 1.

Point 6: Line 134: more information about the NPP-VIIRS is needed e.g., spatial resolution. How it was incorporated in the analysis.

Response 6: The spatial resolution of the NPP-VIIRS and how it was incorporated into the analysis is described in lines 176-178.

Point 7: Section 3.1: add the equations of the spectral indices.

Response 7: The equations for NDBI, NDWI and NDBI spectral indices have been added in Section 3.1.

Point 8: Figure 8: add the full name of the classes with the letters such as:  Residential land (R), and so on…

Response 8: Figure 8 and Figure 9: Added full name of category with letters.

Point 9: Table 3: add the full name of the classes in the columns and rows instead of letters.

Response 9: Table 3 and Table 4: Added the full name of the classes in the columns and rows.

Point 10: Lines 390-402: should be moved to the Methodology section.

Response 10: Section 4.3.2 has been revised to move the section dealing with accuracy verification formulae to section 3.5 of the methodology section.

Point 11: Section 4.4.1 and further sections: Please write the full name of the UFZ within the text instead of using the letters coding. Readers will not memorize what these letters mean, and they should not keep referring to Table 1.

Response 11: We have revised the letter-coded content within section 4.4.1 and other sections to include their full names and the type of urban functional area they represent, so that readers do not have to refer to Table 1 all the time. Also, we did not use the full name of UFZs but UFZs, as we saw in the article [1] that they used "UFZs" instead of "urban functional zones" throughout, so we thought it appropriate to use the UFZ alphabetic code here.

[1]Du Shouhang,Du Shihong,Liu Bo & Zhang Xiuyuan.(2021).Mapping large-scale and fine-grained urban functional zones from VHR images using a multi-scale semantic segmentation network and object based approach. Remote Sensing of Environment. doi:10.1016/J.RSE.2021.112480.

Also, due to an oversight on our part, some of the T-codes in the article were not changed back to S-codes (including Figures 9 and 10). We have corrected this and apologize for this.

Reviewer 3 Report

This study aims to identify urban functional areas and spatial structure analysis in Zhengzhou, China. The paper is well-written, clear, concise yet complete in structure.  However, there are also multiple major problems.

1. There are two key steps to identify urban functional areas. First, the boundary of urban functional areas should be accurately delineared. Second, the accurate classification of each area is needed. However, authors pay more attention to the second part. They ignore the importance of the boundary of urban functional areas. I found that the roads fail to represent in the experimental results.

2. The used the FL-lightGBM to identify urban functional areas. However, there are many hyperparameters in it. How to fine-tune the algorithm to get the best performance should clarify as well.

3. I cannot follow how to convert the object-based classification results into urban functional areas. They should clarify how to handle the boundary inconsistency between objects and urban functional areas.

4. The discussion and conclusion parts is too short. Expand.

5. Focus on the advantages/disadvantages of the study and results.

6. At the end of the manuscript, explain the implications and future works considering the outputs of the current study.

Author Response

Dear Reviewer,

We wish to thank you for your insightful comments. These have greatly helped us to improve the quality of our manuscript.

In accordance with your comments, we have made the following main changes:

Point 1: There are two key steps to identify urban functional areas. First, the boundary of urban functional areas should be accurately delineared. Second, the accurate classification of each area is needed. However, authors pay more attention to the second part. They ignore the importance of the boundary of urban functional areas. I found that the roads fail to represent in the experimental results.

Response 1: Regarding the first issue, since roads are the most important boundary line for different UFZs, we used roads combined with elements such as rivers to segment the study units in order to determine the urban functional area boundary line for each unit parcel, as specified in section 3.3. Because we used the road centerline as the true urban functional zone boundary, and subsumed part of the road area into the unit parcel, the road component is not reflected in the results.

Point 2: The used the FL-lightGBM to identify urban functional areas. However, there are many hyperparameters in it. How to fine-tune the algorithm to get the best performance should clarify as well.

Response 2: The relevant hyperparameters of the model (FL-lightGBM) are described in section 4.2.1. And an additional image was also added to modify Figure 5.

Point 3: I cannot follow how to convert the object-based classification results into urban functional areas. They should clarify how to handle the boundary inconsistency between objects and urban functional areas.

Response 3: Based on the classification of the objects, we determine the type of urban functional area at the block scale by counting the proportion of the area of different urban functional areas within the block unit (i.e. the method described in section 3.4). With regard to the inconsistency of boundaries between objects and urban functional areas, the object units were partitioned twice using the block plot units obtained in section 3.3 to ensure that an object was within a block unit. We also explain this in section 4.3.1.

Point 4: The discussion and conclusion parts is too short. Expand.

Response 4: We have expanded the discussion and conclusion sections as appropriate.

Point 5: Focus on the advantages/disadvantages of the study and results.

Response 5: The advantages/disadvantages of the study and the results are discussed and analyzed accordingly.

Point 6: At the end of the manuscript, explain the implications and future works considering the outputs of the current study.

Response 6: The significance of the study and perspectives for future work are added in the final part of the conclusions in section 6. It has also been appropriately supplemented in the discussion section of section 5.

Round 2

Reviewer 3 Report

Authors addressed most of my concerns; however, the road networks should be added in the results. They can use the central line of roads to get the actual road areas and add in the results. 

Author Response

Thank you again for your further suggestions for our articles. We have made the following changes in response to your suggestions.

Point: Authors addressed most of my concerns; however, the road networks should be added in the results. They can use the central line of roads to get the actual road areas and add in the results.

Response: We have added the road network to the results, modifying Figures 8, 9 and 10. The different widths are given to the different classes of roads and are explained in lines 166-169.

Round 3

Reviewer 3 Report

It's ready for publication.

Author Response

Thank you very much for recognizing our work.